# Long-Term Outcomes of the Dietary Approaches to Stop Hypertension (DASH) Intervention in Nonobstructive Coronary Artery Disease: Follow-Up of the DISCO-CT Study

**DOI:** 10.3390/nu17152565

**Published:** 2025-08-06

**Authors:** Magdalena Makarewicz-Wujec, Jan Henzel, Cezary Kępka, Mariusz Kruk, Barbara Jakubczak, Aleksandra Wróbel, Rafał Dąbrowski, Zofia Dzielińska, Marcin Demkow, Edyta Czepielewska, Agnieszka Filipek

**Affiliations:** 1Institute of Pharmaceutical Care, VIZJA University, Okopowa Street 59, 01-043 Warsaw, Poland; e.czepielewska@vizja.pl; 2Department of Coronary and Structural Heart Diseases, National Institute of Cardiology, 42 Alpejska Street, 04-628 Warsaw, Poland; ckepka@ikard.pl (C.K.); mkruk@ikard.pl (M.K.); zdzielinska@ikard.pl (Z.D.); mdemkow@ikard.pl (M.D.); 3Medical Diagnostic Laboratory, National Institute of Cardiology, 42 Alpejska Street, 04-628 Warsaw, Poland; bjakubczak@ikard.pl (B.J.); awrobel@ikard.pl (A.W.); 4Department of Coronary Artery Disease and Cardiac Rehabilitation, National Institute of Cardiology, 42 Alpejska Street, 04-628 Warsaw, Poland; rdabrowski@ikard.pl; 5Department of Pharmaceutical Biology, Medical University of Warsaw, Banacha 1, 02-097 Warsaw, Poland; agnieszka.filipek@wum.edu.pl

**Keywords:** coronary artery diseases, DASH diet, chemokines CXCL4 and RANTES

## Abstract

In the original randomised Dietary Intervention to Stop Coronary Atherosclerosis (DISCO-CT) trial, a 12-month Dietary Approaches to Stop Hypertension (DASH) project led by dietitians improved cardiovascular and metabolic risk factors and reduced platelet chemokine levels in patients with coronary artery disease (CAD). It is unclear whether these benefits are sustained. Objective: To determine whether the metabolic, inflammatory, and clinical benefits achieved during the DISCO-CT trial are sustained six years after the structured intervention ended. Methods: Ninety-seven adults with non-obstructive CAD confirmed in coronary computed tomography angiography were randomly assigned to receive optimal medical therapy (control group, *n* = 41) or the same therapy combined with intensive DASH counselling (DASH group, *n* = 43). After 301 ± 22 weeks, 84 individuals (87%) who had given consent underwent reassessment of body composition, meal frequency assessment, and biochemical testing (lipids, hs-CRP, CXCL4, RANTES and homocysteine). Major adverse cardiovascular events (MACE) were assessed. Results: During the intervention, the DASH group lost an average of 3.6 ± 4.2 kg and reduced their total body fat by an average of 4.2 ± 4.8 kg, compared to an average loss of 1.1 ± 2.9 kg and a reduction in total body fat of 0.3 ± 4.1 kg in the control group (both *p* < 0.01). Six years later, most of the lost body weight and fat tissue had been regained, and there was a sharp increase in visceral fat area in both groups (*p* < 0.0001). CXCL4 decreased by 4.3 ± 3.0 ng/mL during the intervention and remained lower than baseline values; in contrast, in the control group, it initially increased and then decreased (*p* < 0.001 between groups). LDL cholesterol and hs-CRP levels returned to baseline in both groups but remained below baseline in the DASH group. There was one case of MACE in the DASH group, compared with four cases (including one fatal myocardial infarction) in the control group (*p* = 0.575). Overall adherence to the DASH project increased by 26 points during counselling and then decreased by only four points, remaining higher than in the control group. Conclusions: A one-year DASH project supported by a physician and dietitian resulted in long-term suppression of the proatherogenic chemokine CXCL4 and fewer MACE over six years, despite a decline in adherence and loss of most anthropometric and lipid benefits. It appears that sustained systemic reinforcement of behaviours is necessary to maintain the benefits of lifestyle intervention in CAD.

## 1. Introduction

Despite a 34.9% decline in global cardiovascular mortality between 1990 and 2022, coronary artery disease (CAD) remains the leading cause of death and disability-adjusted life years worldwide [1]. Consequently, international guidelines emphasise primary and secondary prevention through diet, physical activity, and weight management. Studies indicate that dietary interventions can complement medical treatment by further reducing risk, even when drug therapy is optimal. They may also have a beneficial effect on several pathophysiological mechanisms, such as lipids, inflammation, endothelial function and microflora [2,3]. The 2021 European Society of Cardiology (ESC) guidelines recommend predominantly plant-based diets, such as the Mediterranean diet, the Dietary Approaches to Stop Hypertension (DASH) diet, and other plant-forward diets, as first-line nutritional strategies in both preventive settings [4].

Originally designed to lower blood pressure in adults with hypertension, the DASH diet emphasises fruits, vegetables, whole-grain cereals, legumes, and low-fat dairy products, while restricting sodium, red meat, and added sugars. In Appel et al.’s landmark multicentre feeding study, systolic/diastolic blood pressure fell by 5.5/3.0 mmHg within eight weeks, and the subsequent DASH-Sodium trial demonstrated additional benefits when dietary sodium was restricted concurrently [5,6].

Beyond blood pressure control, systematic reviews and meta-analyses consistently demonstrate favourable effects on serum lipids, insulin resistance, and body weight. An umbrella review published in 2019 reported relative risk reductions of 20% for incident cardiovascular disease and 19% for stroke among individuals with the highest DASH adherence [7], while Lari et al. documented significant improvements in total cholesterol (−5.12 mg/dL) and body weight (−1.59 kg) across 54 randomised trials [8]. A meta-analysis of prospective cohorts indicates that a four-point increase in the DASH score is associated with a 5% lower risk of CAD [9], and a 22-year cohort study found that the highest adherence category was associated with a lower incidence of heart failure [10].

However, randomised evidence for hard cardiovascular endpoints remains sparse because few trials extend beyond one year. A 2025 Cochrane review identified only eight studies with a follow-up period of more than 12 months and concluded that reductions in myocardial infarction, stroke, or cardiovascular mortality have yet to be demonstrated [11].

The mechanisms underlying the cardioprotective effects of the DASH diet likely extend beyond sodium reduction and traditional cardiometabolic risk factors [12]. Among plant-based dietary patterns, the DASH diet and the Mediterranean diet appear to elicit the greatest decreases in inflammatory biomarkers, such as C-reactive protein, interleukin-6, and tumour necrosis factor-α [13]. In the single-centre, randomised Dietary Intervention to Stop Coronary Atherosclerosis in Computed Tomography (DISCO-CT) trial (2015–2019), we demonstrated that a 12-month, dietitian-led DASH programme resulted in significant reductions in visceral fat, LDL cholesterol, and circulating platelet chemokines, such as CXCL4 and RANTES, in patients with non-obstructive CAD, compared to optimal medical therapy alone [14]. This is clinically relevant because platelet-derived chemokines, such as CXCL4, promote monocyte recruitment and plaque destabilisation. In vitro studies demonstrate that CXCL4 downregulates the atheroprotective haemoglobin scavenger receptor, CD163, on macrophages [15]. Therefore, CXCL4-induced M4 macrophages are emerging as potential diagnostic and therapeutic targets in human atherosclerosis [16]. Whether the cardiometabolic and anti-inflammatory benefits observed in the DISCO-CT study translate into a long-term clinical advantage remains unknown.

Sustaining dietary changes, including adherence to the DASH diet, over the long term is challenging. Observational data from the PREMIER study indicate that adherence begins to wane as early as 18 months after counselling [17].

Although there is extensive evidence of the short-term benefits of the DASH diet, long-term data on hard outcomes in secondary prevention is scarce. Therefore, the present analysis assesses six-year outcomes after cessation of the structured intervention, focusing on body composition, lipid profile, inflammatory biomarkers, major adverse cardiovascular events (MACE), and long-term adherence to DASH recommendations.

## 2. Materials and Methods

### 2.1. Study Design and Population

This study is a long-term follow-up of the DISCO-CT randomised controlled trial (NCT02571803), which was originally conducted between 2015 and 2019. The original DISCO-CT study enrolled 97 patients with stable CAD confirmed by coronary computed tomography angiography. The patients were randomised into two groups: a control group that received optimal medical treatment and a study group that received the same treatment plus a 12-month lifestyle intervention based on the DASH diet. The inclusion criteria were as follows: coronary artery atherosclerotic lesions confirmed by CTA (defined as maximum luminal narrowing of less than 70% in at least two coronary artery segments); no indication for coronary angiography or revascularisation; age > 18 years; and informed consent to participate in the study. The following exclusion criteria were applied: valvular heart disease or any other condition requiring cardiac surgery; cardiomyopathy; type 2 diabetes; previous coronary artery bypass graft (CABG) surgery; any known genetic disorders affecting the development of atherosclerotic changes; and any factors that may affect the quality and safety of coronary artery CTA. Dietary counselling was implemented at six visits: at baseline and after one, three, six, nine, and 12 months. The primary results of the original trial, including biochemical, anthropometric, and imaging outcomes, have been published previously [14,18,19].

After nearly six years (301.7 ± 21.7 weeks) without a structured clinical follow-up, patients were contacted again and invited to participate in the current study (NCT06031974, 2023–2024). Those who consented underwent the same clinical, dietary, laboratory, anthropometric, and imaging assessments performed at the end of the original DISCO-CT trial. The course of the study and follow-up is shown in Figure 1.

### 2.2. Data Collection Procedures

All follow-up assessments were performed at the same clinical centres using the same protocols as the initial DISCO-CT study to ensure consistency and comparability. Evaluations are included below.

Anthropometric measurements (height, weight, and body composition). Patients’ height and body weight were measured using an electronic BSM370 (Biospace, Seoul, Republic of Korea) device to the nearest 0.1 cm and 0.05 kg, respectively. Body composition was measured with the patient wearing minimal clothing and no shoes using a Body Water Analyzer (Accuniq BC380; SelvasHealthcare, Daejeon, Republic of Korea). The analysis included skeletal muscle mass (SMM), total body fat (TBF), and visceral fat area (VFA), which is defined as the intra-abdominal cross-sectional area of visceral fat, expressed in centimetres squared. Measurements were taken on an empty stomach with at least a 12 h interval from physical activity.

#### 2.2.1. Biochemical Analyses

Blood was collected from fasting patients at the follow-up visit for biochemical tests. Plasma levels of CXCL4 and RANTES were assayed using ELISA kits (Human CXCL4 ELISA Kit and Human RANTES ELISA Kit, both from Biorbyt, Cambridge, UK) according to the manufacturers’ protocols.

#### 2.2.2. Dietary Adherence and Lifestyle Assessment

Long-term adherence to dietary recommendations was estimated based on a food frequency questionnaire (FFQ) with a focus on food group consumption aligned with the DASH diet. Energy and nutrient intake were calculated using the Diet 6 programme based on the results of a 24 h dietary interview. Diet quality and adherence were assessed in both groups using the DASH index according to the methodology described by Guntner et al. [20]. No dietary counselling or interventions were offered between the end of the original study and this follow-up. Physical activity was recorded using self-reported frequency and type of activity, as per ESC guidelines [4].

### 2.3. Statistical Analysis

Changes in body composition and biochemical markers were analysed in relation to the original group assignment. Continuous variables were presented as the arithmetic mean when normally distributed and compared using Student’s *t*-test; otherwise, the median and interquartile range were presented, and a Mann–Whitney U test was applied. Categorical data were compared using the chi-square test or Fisher’s exact test, as appropriate. Correlations were assessed using Spearman’s rank test. Statistical significance was set at *p* < 0.05.

Categorical comparisons of the four-level physical activity score between the DASH and control groups at each time point were assessed using a Pearson chi-square test on 4 × 2 contingency tables. Changes in the ordinal activity distribution within groups were evaluated using the Stuart–Maxwell test for marginal homogeneity (a generalisation of McNemar’s test for more than two categories).

Six-year changes (Δ = follow-up − baseline) were correlated with baseline age using Spearman’s test, a distribution-free measure chosen because several Δ-distributions were skewed. To adjust for study allocation, each Δ outcome was regressed on age and the ARMS indicator in a multivariable linear model (Δ = β_0_ + β_1_Age + β_2_ARMS + ε). Effect estimates (β_1_) are reported with 95% confidence intervals, and a two-sided *p*-value of less than 0.05 was considered significant. All analyses were conducted using IBM SPSS Statistics 19.

### 2.4. Ethics

The follow-up study protocol was approved by the Bioethics Committee of the National Institute of Cardiology in Warsaw (IK.NPIA.0021.19.2018/23). All participants provided renewed informed consent prior to participation in the follow-up.

## 3. Results

During the 12-month intervention, no myocardial infarctions, strokes, or other MACEs were recorded in either group. Over the subsequent follow-up, one death due to myocardial infarction and three additional MACE occurred in the control group, compared to one MACE (non-fatal subarachnoid haemorrhage) in the DASH group (Table 1).

### 3.1. Body Weight and Composition

At baseline, the two groups were comparable in terms of body mass index (BMI) (DASH: 30.09 ± 3.65 kg/m^2^ vs. control: 29.09 ± 3.83 kg/m^2^; *p* = 0.12) and all other anthropometric measures (all *p* > 0.10).

After 12 months, the DASH group lost significantly more body weight (3.62 ± 4.20 kg vs. 1.06 ± 2.86 kg; *p* = 0.002) and showed a greater decrease in BMI (−1.24 ± 1.36 kg/m^2^ vs. 0.64 ± 3.95 kg/m^2^; *p* = 0.006). They also experienced a greater reduction in total body fat (TBF: −4.18 ± 4.78 kg vs. −0.30 ± 4.09 kg; *p* < 0.001) and a modest increase in skeletal muscle mass (ΔSMM = 0.26 kg), while the control group lost an average of 1.00 kg (*p* = 0.018). VFA decreased in both groups (−32.2 ± 41.0 vs. −20.9 ± 33.0 cm^2^), and there was no significant difference between the groups (*p* = 0.17).

Over the six years following DISCO-CT, both groups regained weight and fat mass. The magnitude of regain did not differ significantly except for BMI, which increased slightly more in former DASH participants (0.56 ± 1.80 vs. −1.04 ± 3.92 kg/m^2^; *p* = 0.022). No between-group differences were detected for later changes in TBF, VFA, SMM, or total body water (all *p* > 0.10). Nevertheless, VFA rose significantly in both groups, and after six years, it was higher than baseline in both groups (control group: 64% increase; DASH group: 53% increase). Detailed results on body composition are presented in Table 2.

### 3.2. Dietary Intake and DASH Adherence

At month 12, improvements were seen in all DASH-index components in the intervention arm (all *p* < 0.001). This resulted in an increase of 25.97 ± 3.33 points in the overall DASH score (*p* < 0.0001), which was markedly greater than the increase seen in the control group (4.81 ± 0.23). After six years, adherence waned and the DASH score fell by −4.02 ± 8.60 points (*p* = 0.004); however, it remained superior to the control arm for grains, vegetables, fruits, meats, dairy, and the overall index (Table 3). No significant between-group differences were found for nuts, fats, or sweets.

Energy intake decreased by −191.68 ± 390.49 kcal/d in the DASH group at 12 months (*p* < 0.001) and then increased, not significantly, by 146.45 ± 535.11 kcal/d, between months 12 and 60 (*p* = 0.084). The control group showed no change in year one and a modest reduction thereafter (72.21 ± 363.52 kcal/d; *p* = 0.049). The proportion of energy from total fat decreased in the DASH group by −4.22 ± 5.36% at month 12 (*p* < 0.001) and remained stable; no changes occurred in the control group. Saturated fat intake increased significantly in both groups during the follow-up period, with a larger absolute increase in the intervention group (3.14 ± 5.91 g; *p* = 0.002). No other nutrient changes reached statistical significance (Table 3). Restrictions related to the COVID-19 pandemic were indicated as a factor with an adverse impact on eating habits by 7 (8.3%) patients.

### 3.3. Physical Activity

By the end of the original study, 83.7% of the DASH participants reported regular physical activity, compared to 48.8% of the control group. Only 4.7% of the DASH participants versus 19.5% of the control group fell into the “none/seldom” categories (χ^2^ *p* = 0.008). During long-term follow-up, activity declined in both groups. Among the control group, regular activity decreased to 31.7%, while the none/seldom category increased to 48.7% (Stuart–Maxwell test, *p* = 0.020). The DASH group experienced a more significant shift: regular activity decreased to 41.9%, while the none/seldom category increased to 37.2% (*p* = 0.003). By 2023, the activity distributions no longer differed between the two groups (χ^2^ *p* = 0.576). Restrictions related to the COVID-19 pandemic were indicated as a factor with an adverse impact on physical activity by 31 (36.9%) patients.

### 3.4. Diet and Biochemical Markers

#### 3.4.1. Lipid Profile

During the 12-month intervention, the DASH group experienced a clinically significant decrease in total cholesterol (−22.4 ± 36.1 mg/dL; *p* < 0.001) and LDL-C (−18.2 ± 33.1 mg/dL; *p* = 0.001). In contrast, the control group showed smaller, non-significant changes. These improvements attenuated over the ensuing six years. Total cholesterol in the DASH group returned to baseline, and LDL-C increased by 12.0 ± 42.9 mg/dL. This yielded no difference between groups at follow-up. HDL-C increased modestly in both groups, and triglycerides showed a temporary increase in the DASH group, though there were no significant differences between groups at any time point. Non-HDL-C decreased more in the intervention group during the original trial yet converged with control values by year six (Table 4).

#### 3.4.2. Inflammatory Markers

High-sensitivity C-reactive protein (hs-CRP) decreased by 0.09 ± 0.22 mg/L in the DASH group during the intervention (*p* = 0.010), but then returned to baseline levels. The control group showed no significant changes, and there were no significant differences between the groups during the observation period. The level of CXCL4 chemokine decreased significantly in the DASH group (−4.34 ± 3.02 ng/mL; *p* < 0.001), while it increased in the control group (2.19 ± 5.13 ng/mL; *p* = 0.010). During follow-up, CXCL4 levels decreased in the control group and increased modestly in the DASH group. Both changes within groups were significant (*p* < 0.01), but the magnitude of change differed between groups (*p* < 0.001). Unfortunately, after 6 years, there were no longer any significant differences between the groups (*p* = 0.54). Similarly, despite large differences at the end of the intervention, RANTES levels were almost equal after 6 years. However, it should be noted that despite these unfavourable changes after six years, the values for patients in the DASH group were lower than the baseline values for both chemokines (CXCL4, *p* < 0.001; RANTES, *p* = 0.260), while the control group had significantly higher RANTES levels (*p* < 0.001) and similar CXCL4 levels (*p* = 0.116) to baseline values.

#### 3.4.3. Homocysteine

Plasma homocysteine fell by 2.1 ± 8.3 µmol/L in the DASH group and by 0.9 ± 2.7 µmol/L in the control group during the intervention. However, neither reduction remained significant after six years, and no intergroup differences emerged.

### 3.5. Correlation Analysis

#### 3.5.1. Correlation Between Dietary Changes and Body Composition

A statistical analysis revealed that the six-year change in the overall DASH index was inversely associated with changes in VFA (r = −0.272, *p* = 0.013) and TBF (r = −0.380, *p* < 0.001). The Sweets Index change correlated negatively with TBF (r = −0.463, *p* < 0.001), while the Fats Index change showed an inverse relation with VFA (r = −0.222, *p* = 0.044). Additionally, the six-year change in daily energy intake positively correlated with changes in body weight (r = 0.280, *p* = 0.011), total body fat percentage (r = 0.304, *p* = 0.005), and VFA (r = 0.278, *p* = 0.012). No other significant correlations were observed.

#### 3.5.2. Correlations Between Dietary Changes and Biochemical Parameters

Dietary fibre showed the following significant relationships: increases in fibre intake were inversely associated with hs-CRP levels after six years (r = −0.281, *p* = 0.020) and with the six-year change in hs-CRP (r = −0.263, *p* = 0.030). Fibre intake was not related to RANTES or CXCL4, and no other dietary components correlated with lipid or inflammatory markers. Significant correlations were found between the Nuts Index and the change in RANTES concentration (r = −0.240, *p* = 0.028).

#### 3.5.3. Inter-Marker Correlations

Overall, Δhs-CRP was unrelated to ΔRANTES (*n* = 79; r = 0.002, *p* = 0.983), but it correlated moderately with ΔCXCL4 (*n* = 78; r = 0.375, *p* < 0.001). Subgroup analysis revealed no association between hs-CRP and RANTES in either group. A strong positive correlation between hs-CRP and CXCL4 was present in the control group (r = 0.433, *p* = 0.006), but not in the DASH group (r = −0.058, *p* = 0.727). Fisher’s Z test confirmed a significant difference between the two groups’ correlation coefficients (Z = 2.213, *p* = 0.027).

#### 3.5.4. The Impact of Baseline Age on Changes to Parameters

There was no significant correlation between baseline age and six-year changes in LDL-C, CXCL4, RANTES, VFA, BMI or TBF (Spearman r ≤ 0.16; *p* ≥ 0.15), and the age coefficients from linear models adjusted for study arm were also non-significant (β range = −0.47 to 0.18; *p* ≥ 0.08), except for VFA. VFA increased more rapidly in older participants, with each additional year of baseline age being associated with an increase of 1.85 cm^2^ over six years (β = 1.85 cm^2^; *p* = 0.037). These results suggest that, aside from VFA, age did not significantly impact the longitudinal changes observed in the studied cardiometabolic markers (Appendix A).

## 4. Discussion

To our knowledge, this is the first study to assess the six-year sustainability of the effects of the DASH diet intervention on lipid parameters, body composition, and inflammation. The six-year follow-up of the DISCO-CT trial shows that an intensive, clinician-supported DASH intervention for patients with non-obstructive CAD is beneficial, but these benefits are unfortunately lost after the supervised phase. Long-term observational data on hard endpoints is lacking in the literature. A 2025 Cochrane review of DASH for primary and secondary prevention revealed that most studies focused on risk factors rather than clinical outcomes and had follow-up periods ranging from 16 weeks to 18 months [9]. During our follow-up, there was one MACE in the DASH group, compared to four events, including one fatal myocardial infarction, in the control group (*p* = 0.575). However, the small study sample size prevents us from drawing firm conclusions.

Changes in body composition appear to be a key mediator of cardiometabolic improvement. During the 12-month intervention, the DASH group achieved significantly greater reductions in weight, BMI, and TBF mass than the control group. These results are consistent with meta-analyses linking higher adherence to DASH recommendations with lower obesity and cardiometabolic risk [12,21]. Although the participants regained some weight and fat over the following six years, VFA increased by 53% from baseline in the intervention group compared to 64% in the control group. VFA is considered an independent cardiovascular risk factor given its proven role in the development of atherogenic dyslipidaemia, insulin resistance, and low-grade chronic inflammation [22]. Therefore, even a modest, sustained attenuation of VFA expansion may benefit long-term prognosis. It is important to note that a decrease in overall DASH index scores was associated with an increase in TBF and VFA. Decreases in DASH scores for sweetness were associated with increases in body fat, while decreases in scores for fat were associated with increases in VFA. However, it should be noted that the increase in VFA can be partly explained by the influence of age, as demonstrated in our statistical analysis.

The observation period (2017–2023) coincided with the onset of the COVID-19 global pandemic, during which Poland implemented strict stay-at-home measures. National surveys revealed increased consumption of total and saturated fats, as well as sugars. Additionally, the percentage of adults classifying their physical activity level as low increased to 50.8% [23,24]. Similar trends likely affected our cohort. Energy intake increased by ~150 kcal/day in the intervention group (*p* = 0.084) but remained unchanged in the control group. Saturated fat intake increased in both groups. These dietary changes, coupled with a decline in physical activity, were associated with elevated body weight, VFA, and TBF. Positive correlations between changes in energy intake and changes in body weight (r = 0.280, *p* = 0.011), TBF (r = 0.304, *p* = 0.005), and VFA (r = 0.278, *p* = 0.012) confirm this association.

Short-term reductions in total and LDL-C mirrored earlier findings in primary-prevention settings [8,12] but largely dissipated once structured support ceased, as did the fall in hs-CRP. Overall, however, LDL-C, hs-CRP, homocysteine, RANTES, CXCL4 and non-HDL-C remained below baseline in the DASH arm, whereas hs-CRP, homocysteine and RANTES rose above baseline in controls. Particularly striking was the decline in CXCL4 during the intervention, which remained ~30% lower than in controls six years later. While Soltani et al.’s meta-analysis confirmed modest DASH-induced reductions in hs-CRP over a shorter follow-up period [25], chemokines are rarely assessed in dietary trials. The durable suppression of CXCL4 is therefore novel and potentially significant, given its role in monocyte recruitment and plaque destabilisation. CXCL4 induces the polarisation of monocytes into the M4 phenotype, which predominates in unstable human arterial plaques [16]. Deletion of the Pf4 gene in the ApoE^−/−^ model results in a significant decrease in the size of atherosclerotic lesions, which confirms the causal nature of this axis [26]. The absence of a corresponding long-term decrease in hs-CRP highlights the importance of evaluating multiple inflammatory pathways instead of focusing on a single acute-phase protein.

A significant decrease in RANTES levels was observed in both groups during the follow-up period. Studies in vitro and in vivo have shown that the chemokine CXCL4 forms heterodimers with RANTES, strongly enhancing monocyte adhesion to activated endothelium and accelerating atherosclerotic plaque formation [27]. Although RANTES and CXCL4 changes were correlated during the intervention, this association disappeared after six years. The final RANTES levels (40.24 ± 19.98 ng/mL in the DASH group and 41.02 ng/mL in the control group) were comparable to levels observed in patients with obstructive CAD [28]. Correlation analysis revealed that the only dietary factor associated with RANTES change was nut consumption. The effect of nut consumption on inflammation remains unconfirmed. A systematic review by Mazidi et al. suggests that nut consumption significantly reduces leptin, though it has no significant effect on C-reactive protein (CRP), interleukin-6 (IL-6), adiponectin, interleukin-10 (IL-10), or tumour necrosis factor-alpha (TNF-α) [29]. Further studies are needed to confirm the association with RANTES levels.

Correlation analyses highlight the importance of dietary fibre. Higher fibre intake during the six-year follow-up period was negatively correlated with hs-CRP, reflecting recent findings from the NHANES observational study that higher fibre intake is associated with lower systemic inflammation [30]. Although fibre intake remained higher in the intervention group, consistent with better DASH index scores for vegetables and whole grain products, it declined significantly after the intervention phase. This indicates difficulty in maintaining optimal intake.

Maintaining adherence proved to be a major challenge for patients. The average DASH index score increased by 26 points by month 12 but dropped by 4 points by year 6. Regular physical activity, previously reported by 84% of patients, was reported by only 42% of patients. These declines confirm earlier reports indicating that only 15–50% of cardiac rehabilitation graduates maintain regular exercise six months after completing the programme [31]. Systematic support, such as regular contact with a dietitian, digital tools, and behavioural incentives, may therefore be crucial. New hybrid m-health strategies combining online education, self-monitoring, and gamification to promote long-term dietary adherence in cardiovascular patients may be effective motivators after rehabilitation ends [32]. Another promising model is remote rehabilitation delivered via wearable devices and short coaching sessions [33].

## 5. Conclusions

In conclusion, a behavioural intervention based on the DASH diet model produced short-term improvements in obesity, lipid profile, and systemic inflammation. However, without continued specialist support, these improvements were largely lost after six years. The loss of benefit due to suboptimal diet and decreased physical activity highlights the need for sustained patient support strategies in clinical practice and in long-term interventions. Additionally, investigating chemokines such as CXCL4 as prognostic markers that go beyond conventional risk factors is crucial. Our results suggest that moderate but consistent adherence to the DASH diet, accompanied by systematic behavioural support to maintain healthy habits and clinical benefits, is a valuable supplement to pharmacotherapy.

Limitations: The study’s limitations include its small sample size, single-centre recruitment, reliance on self-reported diet and activity, and the observational nature of the post-intervention period. The results should be treated with caution due to the small sample size. Further multicentre studies involving larger sample sizes are required to confirm the obtained results.

## Figures and Tables

**Figure 1 nutrients-17-02565-f001:**
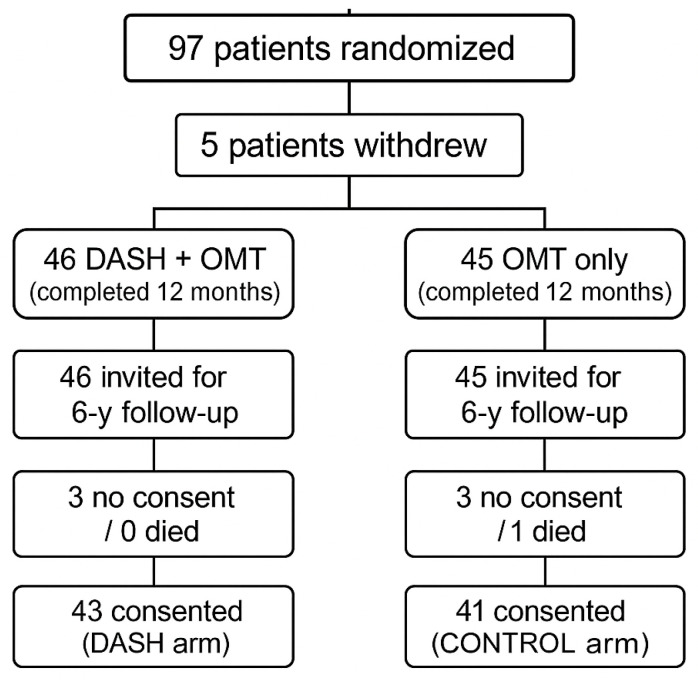
Flow diagram of DISCO-CT intervention and follow-up.

**Table 1 nutrients-17-02565-t001:** Characteristics of the study groups and concomitant medication at baseline and follow-up.

	Baseline		Final		Follow-Up		*p*-Value
	DASH *n* = 43	Control *n* = 41	DASH *n* = 43	Control *n* = 41	DASH *n* = 43	Control *n* = 41	
Age, years (mean ± SD)	58.8 ± 7.7	59.9 ± 7.8	60.0 ± 7.8	61.7 ± 4.9	65.81 ± 7.79	67.54 ± 7.45	0.304
Women, *n* (%)	14 (32.6%)	18 (43.9%)	14 (32.6%)	18 (43.9%)	14 (32.6%)	18 (43.9%)	0.369
BMI, kg·m^−2^	30.0 ± 3.65	29.0 ± 3.83	28.8 ± 3.36	29.79 ± 4.42	29.36 ± 3.50	28.48 ± 3.52	0.260
Current smoker, *n* (%)	5 (11.6%)	8 (19.5%)	3 (7.0%)	5 (12.2%)	3 (7.0%)	6 (14.6%)	0.257
Physical activity, *n* (%)							
Regular	22 (51.2%)	12 (29.3%)	36 (83.7%)	20 (48.7%)	20 (46.5%)	13 (31.7%)	0.473
Irregular	12 (27.9%)	19 (46.3%)	5 (11.6%)	13 (30.2%)	14 (32.5%)	19(19.5%)	0.425
Rare/seldom	9 (20.9%)	8 (19.5%)	2 (4.7%)	8 (19.5%)	8 (18.6%)	6 (48.7%)	0.640
Hypertension, *n* (%)	39 (90.7%)	35 (85.4%)	33 (7.7%)	32 (78.0%)	33 (76.7%)	32 (78.7%)	>0.999
MI, *n* (%) *	n/a	n/a	0	0	0 (0.0%)	2 (4.4%)	0.981
Stroke, *n* (%) *	n/a	n/a	0	0	1 (2.3%)	1 (2.2%)	>0.999
MACE, *n* (%) *	n/a	n/a	0	0	1 (2.3%)	4 (8.9%)	0.575
PCI, *n* (%) *	n/a	n/a	0	0	1 (2.3%)	2 (4.4%)	0.966
CABG, *n* (%) *	n/a	n/a	0	0	0	1 (4.4%)	0.981
Diabetes mellitus type 2, *n* (%)	1 (2.3%)	0 (0.0%)	1 (2.3%)	0 (0.0%)	6 (14.0%)	2 (4.9%)	0.26
Prediabetes, *n* (%)	4 (9.3%)	3 (7.3%)	4 (9.3%)	3 (7.3%)	9 (20.9%)	14 (34.1%)	0.409
Statin therapy, *n* (%)	28 (65.1%)	29 (70.7%)	34 (79.1%)	32 (78.0%)	30 (69.8%)	30 (73.2%)	0.811
High-intensity statin, *n* (%)	7 (16.3%)	7 (17.1%)	9 (20.9%)	8 (19.5%)	12 (27.9%)	10 (2.0%)	0.804
Other lipid-lowering drugs	3 (7.0%)	3 (7.3%)	2 (4.7%)	2 (4.9%)	4 (9.3%)	8 (19.5%)	0.222
Calcium-channel blocker, *n* (%)	12 (27.9%)	12 (29.3%)	11 (25.6%)	14 (34.1%)	12 (27.9%)	19 (46.3%)	0.113
β-blocker, *n* (%)	23 (53.5%)	26 (63.4%)	24 (55.8%)	26 (63.4%)	22 (51.2%)	26 (63.4%)	0.376
ACE-I/ARB, *n* (%)	27 (62.8%)	31 (75.6%)	26 (60.5%)	31 (75.6%)	25 (58.1%)	29 (70.0%)	0.260
ASA, *n* (%)	25 (58.1%)	24 (58.5%)	30 (69.8%)	29 (70.7%)	27 (62.8%)	22 (53.7%)	0.432

* values are for the entire group that completed the 12-month intervention (*n* = 91); *p* indicates the difference between the follow-up values; BMI, body mass index; CABG, coronary artery bypass grafting; PCI, percutaneous coronary intervention; MACE, major adverse cardiovascular events; MI, myocardial infarction; ARB, angiotensin receptor blocker; ACE-I, angiotensin-converting enzyme inhibitor; ASA, acetylsalicylic acid; n/a, not applicable.

**Table 2 nutrients-17-02565-t002:** Change in anthropometric parameters and body composition during DISCO-CT trial and follow-up period.

Parameter	Group	Baseline	Final	6-Year Follow-Up	Δ (Mean ± SD)	Δ* (Mean ± SD	*p*-Value	*p*-Value *	*p*-Value Δ	*p*-Value Δ*
Weight, kg	DASH	85.94 ± 14.89	82.32 ± 13.37	84.01 ± 13.73	−3.62 ± 4.20	1.69 ± 5.14	**<0.001**	0.036	**0.002**	0.068
	Control	83.41 ± 15.38	82.35 ± 15.46	82.04 ± 14.43	−1.06 ± 2.86	−0.31 ± 4.80	**0.023**	0.679		
BMI, kg/m^2^	DASH	30.09 ± 3.65	28.86 ± 3.36	29.36 ± 3.51	−1.24 ± 1.36	0.56 ± 1.80	<0.001	0.052	**0.006**	**0.022**
	Control	29.09 ± 3.83	29.73 ± 4.42	28.48 ± 3.53	0.64 ± 3.95	−1.04 ± 3.92	0.308	0.100		
TBF, kg	DASH	29.20 ± 9.03	25.01 ± 7.52	28.34 ± 6.83	−4.18 ± 4.78	3.33 ± 5.55	<0.001	<0.001	**<0.001**	0.111
	Control	26.68 ± 7.90	26.98 ± 8.16	28.09 ± 8.59	0.30 ± 4.09	1.22 ± 6.31	0.644	0.230		
SMM, kg	DASH	31.65 ± 7.10	31.91 ± 6.74	31.42 ± 6.21	0.26 ± 2.63	−0.48 ± 1.91	0.520	0.104	**0.018**	0.177
	Control	31.50 ± 7.15	30.51 ± 7.35	31.91 ± 10.28	−1.00 ± 2.10	1.33 ± 8.16	**0.004**	0.309		
VFA, cm^2^	DASH	114.54 ± 44.88	82.32 ± 13.37	175.81 ± 74.33	−32.22 ± 41.03	93.49 ± 69.42	**<0.001**	**<0.001**	0.167	0.691
	Control	103.26 ± 37.47	82.35 ± 15.46	169.93 ± 72.31	−20.91 ± 33.02	87.53 ± 66.85	**<0.001**	**<0.001**		
Total Body Water, L	DASH	41.60 ± 8.40	41.57 ± 8.18	41.29 ± 8.14	−0.01 ± 1.83	−0.29 ± 12.37	0.986	0.880	0.145	0.950
	Control	41.23 ± 8.63	40.37 ± 8.61	40.37 ± 8.74	−0.86 ± 3.16	−0.11 ± 13.86	0.090	0.962		

BMI, body mass index; TBF, total body fat; SMM, skeletal muscle mass; VFA, visceral fat area; Δ, change between baseline and final DISCO-CT; Δ*: change between final DISCO-CT and follow-up; *p* indicates differences within the group (baseline-final) (paired samples *t*-test). *p* * Indicates differences within the group (final follow-up) (paired samples *t*-test); *p* Δ indicates the difference in change (baseline-final) between the two groups (independent samples *t*-test); *p* Δ* indicates the difference in change (final follow-up) between the two groups (independent samples *t*-test). *p* < 0.05 marked in bold.

**Table 3 nutrients-17-02565-t003:** Daily intake of selected nutrients and DASH index values in both study groups during the DISCO-CT trial and follow-up period.

	DASH *n* = 43			*p*-Value	*p*-Value *	Control *n* = 41			*p*-Value	*p*-Value *	*p*-Value Δ
	Baseline	Final	Follow-Up			Baseline	Final	Follow-Up			
Daily nutrients intake											
Energy, kcal	1998.8 ± 669.5	1654.4 ± 442	1804.6 ± 551.9	<0.001	0.084	1898 ± 669.6	2000.2 ± 745	1936 ± 586.6	0.456	0.490	0.094
Fat energy, %	32.3 ± 6.9	28.7 ± 5.8	28.74 ± 5.58	0.01	0.917	31.3 ± 8.4	32.3 ± 8.15	29.63 ± 5.05	0.815	0.072	0.161
SAFA, g	21.4 ± 9.7	18.2 ± 6.2	21.28 ± 9.64	0.042	**0.002**	22.8 ± 12.9	24.17 ± 8.84	27.54 ± 12.58	0.605	**0.004**	0.85
Vit. A, µg	604.8 ± 35.7	633.3 ± 494.4	645.44 ± 509.63	0.809	0.003	1002.18 ± 206.8	714.79 ± 285.1	700.2 ± 283.6	0.146	0.749	0.479
Vit. E, mg	7.95 ± 4.62	12.62 ± 7.03	14.43 ± 13.2	**<0.001**	0.352	10.07 ± 5.34	11.27 ± 7.24	10.74 ± 7.12	0.417	0.267	0.231
Folic Acid, µg	192.1 ± 0.4	253.27 ± 13.2	242.5 ± 114.9	**0.003**	0.382	253.58 ± 58.79	244.56 ± 79.21	241.26 ± 79.39	0.76	0.556	0.619
Vit. C, mg	72.31 ± 8.9	93.4 ± 63.4	96.91 ± 78.82	**0.021**	0.863	66.05 ± 34.03	83.55 ± 61.58	76.63 ± 68.59	0.122	0.306	0.592
Fibre, g	21.44 ± 0.63	26.62 ± 7.36	24.96 ± 9.97	<0.001	0.25	22.99 ± 9.42	22.10 ± 9.57	23.32 ± 11.98	0.595	0.827	0.472
DASH index components											
Grains	5.30 ± 2.44	8.26 ± 1.62	7.81 ± 1.80	<0.001	**0.024**	5.49 ± 2.19	5.51 ± 2.20	5.29 ± 2.15	0.173	0.095	0.334
Vegetables	4.53 ± 1.97	7.07 ± 1.88	6.77 ± 1.91	**<0.001**	0.217	4.63 ± 1.81	5.20 ± 1.94	5.15 ± 1.90	0.026	0.847	0.469
Fruits	4.16 ± 2.35	6.42 ± 3.09	5.72 ± 2.64	**<0.001**	**0.007**	3.90 ± 2.45	4.12 ± 2.33	4.56 ± 2.16	0.071	**0.018**	**<0.001**
Diary	5.72 ± 2.71	7.42 ± 2.59	6.98 ± 2.60	**0.001**	**0.024**	5.63 ± 2.67	6.05 ± 2.60	5.63 ± 2.42	0.058	**0.011**	0.912
Meat	4.16 ± 3.34	8.60 ± 1.93	8.47 ± 2.09	**<0.0001**	0.183	4.54 ± 3.29	5.22 ± 3.12	4.98 ± 2.86	**0.001**	0.133	0.584
Nuts	4.16 ± 3.34	8.60 ± 1.93	8.47 ± 2.09	**<0.0001**	0.183	1.07 ± 2.44	1.71 ± 2.62	1.80 ± 2.63	**0.003**	0.16	0.863
Fat	6.00 ± 3.77	8.65 ± 1.78	8.33 ± 2.04	**<0.001**	0.333	6.02 ± 3.64	6.29 ± 3.56	6.17 ± 3.60	0.147	0.168	0.605
Sweets	3.07 ± 3.78	7.49 ± 3.19	6.65 ± 3.37	**<0.0001**	0.21	6.63 ± 5.65	4.02 ± 2.20	4.39 ± 5.65	**<0.0001**	0.803	0.337
DASH index	33.70 ± 4.64	59.67 ± 8.92	55.65 ± 9.99	**0.0001**	**0.004**	34.93 ± 3.28	38.12 ± 2.20	37.98 ± 11.91	**<0.0001**	0.803	**<0.001**

*p* indicates differences within the group (baseline-final) (paired samples *t*-test). *p* * indicates differences within the group (final follow-up) (paired samples *t*-test); *p* Δ indicates the difference between the change (baseline-final) in the two groups (independent samples *t*-test). *p* < 0.05 marked in bold.

**Table 4 nutrients-17-02565-t004:** Changes in lipid profile, homocysteine and inflammatory markers levels during the DISCO-CT trial and follow-up period.

	Study Group	Baseline	Final	Follow-Up	Change (Δ)	Change (Δ) *	*p*-Value	*p*-Value *	*p*-Value Δ	*p*-Value Δ*
Total cholesterol, mg/dL	DASH	177.68 ± 44.21	167.14 ± 31.97	177.42 ± 44.66	−22.44 ± 36.11	14.58 ± 49.56	**<0.001**	0.06	0.162	0.209
	control	180.14 ± 42.93	165.68 ± 8.62	170.10 ± 36.04	−10.54 ± 40.87	2.96 ± 33.36	0.106	0.573		
LDL-C, mg/dL	DASH	109.27 ± 0.75	91.11 ± 31.84	103.07 ± 39.66	−18.16 ± 33.13	11.96 ± 42.92	**0.001**	0.075	0.27	**0.046**
	control	109.11 ± 40.59	99.70 ± 29.47	95.51 ± 30.62	−9.41 ± 38.67	−4.19 ± 28.91	0.127	0.359		
HDL-C, mg/dL	DASH	54.55 ± 15.32	57.13 ± 16.02	58.72 ± 13.89	2.58 ± 8.52	1.59 ± 11.37	0.053	0.364	0.328	0.823
	control	57.90 ± 14.11	58.77 ± 15.38	60.90 ± 15.84	0.87 ± 7.43	2.13 ± 10.65	0.459	0.208		
Triglycerides, mg/dL	DASH	88.36 ± 16.68	131.76 ± 242.17	114.48 ± 56.12	43.40 ± 140.01	−19.17 ± 50.87	0.048	0.623	0.086	0.859
	control	114.09 ± 1.06	113.94 ± 64.72	101.83 ± 38.39	−0.16 ± 83.23	−12.11 ± 6.00	0.99	0.174		
hs-CRP, mg/L	DASH	0.23 ± 0.24	0.12 ± 0.13	0.21 ± 0.16	−0.09 ± 0.22	0.09 ± 0.14	**0.01**	<0.001	0.188	0.237
	control	0.19 ± 0.17	0.26 ± 0.75	0.20 ± 0.17	0.08 ± 0.78	−0.06 ± 0.75	0.535	0.638		
Homocysteine, umol/L	DASH	14.13 ± 8.83	11.98 ± 2.30	12.24 ± 3.16	−2.09 ± 8.30	0.26 ± 4.04	0.111	0.674	0.394	0.353
	control	12.67 ± 3.60	11.74 ± 3.51	12.90 ± 2.92	−0.93 ± 2.69	1.16 ± 4.58	0.033	0.118		
RANTES, ng/mL	DASH	41.66 ± 22.62	36.94 ± 18.46	40.24 ± 19.98	–4.72 ± 29.23	3.29 ± 27.20	0.244	0.408	**0.048**	0.637
	control	34.60 ± 22.74	40.27 ± 19.85	41.02 ± 19.02	5.67 ± 24.66	0.74 ± 22.44	0.144	0.841		
CXCL4, ng/mL	DASH	12.72 ± 4.25	8.37 ± 2.42	9.85 ± 2.19	−4.34 ± 3.02	1.45 ± 3.24	**<0.001**	**0.008**	**<0.001**	**<0.001**
	control	11.07 ± 4.14	13.33 ± 5.02	10.16 ± 2.47	2.19 ± 5.13	−3.15 ± 5.03	**0.008**	**<0.001**		
Non-HDL-C mg/dl	DASH	130.74 ± 40.41	105.71 ± 35.13	118.70 ± 40.81	−25.03 ± 37.05	12.99 ± 52.69	**<0.001**	0.114	0.108	0.19
	control	119.78 ± 41.76	108.37 ± 31.26	109.20 ± 32.65	−11.41 ± 39.63	0.83 ± 28.52	0.073	0.854		

Δ: change between baseline and final DISCO-CT; Δ*: change between final DISCO-CT and follow-up; *p* indicates differences within the group (baseline-final) (paired samples *t*-test). hs-CRP, high-sensitivity C-reactive protein; HDL-C, high-density lipoprotein cholesterol; LDL, low-density lipoprotein cholesterol; *p* * indicates differences within the group (final follow-up) (paired samples *t*-test); *p* Δ indicates the difference in change (baseline-final) between the two groups (independent samples *t*-test); *p* Δ* indicates the difference in change (final follow-up) between the two groups (independent samples *t*-test). *p* < 0.05 marked in bold.

## Data Availability

The data presented in this study are available on request from the corresponding authors. The data are not publicly available because the study is funded by the Institute of Cardiology in Warsaw, and management approval is required.

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
