# Peer review of "Long-Term Outcomes of the Dietary Approaches to Stop Hypertension (DASH) Intervention in Nonobstructive Coronary Artery Disease: Follow-Up of the DISCO-CT Study"

_nutrients, 2025, doi:10.3390/nu17152565_

Round 1

Reviewer 1 Report

Comments and Suggestions for Authors

Authors have investigated the effect of Dietary interventions for a duration of 6 years to attenuate hypertension and atherosclerosis in the cases of Nonobstructive type of coronary artery disease.

The study seems interesting and worth investigating. I just have a few comments.

  1.  6 ± 4.2 kg, 1.1 ± 2.9 kg, 4.3 ± 3.0 ng/ml in the abstract creates confusion. The values after± signify % or what? Please clarify.
  2.  Inclusion and exclusion can be mentioned. It will clarify the unbiasedness.
  3.  The existing gaps in this research field novelty of the study should be explicitly described.
  4. This can be added at the end of the Introduction section (last paragraph). This will facilitate the novelty of the study.
  5.  The study does not present any underlying molecular mechanisms. Please link it with the core mechanisms of the disease pathology.
  6.  Can the authors explain the superiority of dietary approaches over the conventional
    approved drugs in terms of efficacy and safety? This explanation will help the audience understand the importance of dietary resources and their advantages over the existing therapies.
  7.  The authors have conducted the study between the period 2015 to 2019. I am wondering if the report was submitted after almost 6 years of completing the study. May I know the reason for this gap?
  8. Can the authors report the number of males and females in the study? Is there a possibility of the influence of gender on this dietary therapy?
  9.  Will there be an influence of age on the results, as with an increase and age, there are
    changes in metabolism, immunity, etc., and other factors affecting blood pressure. In this regard, the study lasted for 6 years. How would it affect the results based on age? Would it create any bias in the results? All these points can be discussed.
  10.  What was the rationale for providing the funding information in section 2.5 in the middle of the manuscript before the results? Won’t it be suitable to keep it at the end, preferably after the Conclusion section?

Reviewer 2 Report

Comments and Suggestions for Authors

The manuscript is well written and contains a important of information. However, I have a few comments including:

1'It would be helpful if the authors could elaborate more clearly on the clinical implications of their findings.

2. The number of participants appears to be relatively small for a study of this kind of study.

3. We can not generalize with such a small number of participants. 

4. It would have been more informative if key parameters such as lipid profile, inflammatory markers, and homocysteine levels would have been monitored periodically over time rather than only at a 6-year follow-up.

Same applies to changes in anthropometric measurements and body composition. 
